# Systematic identification of protein combinations mediating chromatin looping

Kai Zhang[1], Nan Li[2,3], Richard I. Ainsworth[2,3] & Wei Wang[1,2,3]

Chromatin looping plays a pivotal role in gene expression and other biological processes through bringing distal regulatory elements into spatial proximity. The formation of chromatin loops is mainly mediated by DNA-binding proteins (DBPs) that bind to the interacting sites and form complexes in three-dimensional (3D) space. Previously, identification of DBP cooperation has been limited to those binding to neighbouring regions in the proximal linear genome (1D cooperation). Here we present the first study that integrates protein ChIP-seq and Hi-C data to systematically identify both the 1D- and 3D-cooperation between DBPs. We develop a new network model that allows identification of cooperation between multiple DBPs and reveals cell-type-specific and -independent regulations. Using this framework, we retrieve many known and previously unknown 3D-cooperations between DBPs in chromosomal loops that may be a key factor in influencing the 3D organization of chromatin.

[1] Graduate Program in Bioinformatics and Systems Biology, University of California, La Jolla, San Diego, California 92093-0359, USA. [2] Department of Chemistry and Biochemistry, University of California, La Jolla, San Diego, California 92093-0359, USA. [3] Department of Cellular and Molecular Medicine, University of California, La Jolla, San Diego, California 92093-0359, USA. Correspondence and requests for materials should be addressed to W.W. (email: wei-wang@ucsd.edu).

The human genome is tightly packaged into chromatin and forms complex structures of which the functional outputs, such as gene expression, depend on local chromatin states and chromatin three-dimensional (3D) organization[1–11]. Chromatin loops are formed to bring distal regulatory elements such as enhancers and their target promoters to spatial proximity. The formation of chromatin loops is mainly regulated by proteins that bind to the 3D interaction sites and form complexes[1]. Previous studies have shown that perturbation of the binding of these proteins could disrupt the loops, which suggests an important role for DNA-binding proteins (DBPs) in genome organization. Mediators of chromatin loops including CTCF, cohesin and several transcription factors (TFs) such as GATA1 and KLF1 have been identified (refs 12–16). Particularly, a recent study has uncovered about 10,000 chromatin loops using kilobase-resolution Hi-C data and discovered that CTCF and cohesin subunits RAD21 and SMC3 are present in the majority of the loops[17].

These studies have shown that the cooperation of multiple DBPs is critical to orchestrate loop formation. However, there still lacks a systematic method to investigate the role of combinatorial regulation between DBPs in chromatin loop formation. Previous studies have focused on identifying DBPs binding to proximal genomic regions[18–21], which is hereinafter referred to as 1D-cooperation. Despite the great insight provided by these studies in revealing the combinatorial regulation of DBPs, they could not detect the cooperation between DBPs binding to distal genomic loci that are localized spatially and form long-range interactions (referred to as 3D-cooperation in this study to be distinct from the 1D-cooperation of DBPs in neighbouring genomic loci). 3D-cooperation of DBPs is key to mediating chromatin looping, either enhancing the existing 3D contacts or creating new ones to bring functional elements, such as enhancers, to their target loci, such as promoters. Despite its importance, no study has thoroughly investigated the DBPs' 3D-cooperation and its relationship to 1D-cooperation.

The ENCODE project has generated hundreds of ChIP-seq data sets to map binding sites of DBPs in multiple cell lines[19,22,23]. Recently, kilobase-resolution Hi-C data were available in two of these cell lines, namely GM12878 and K562 (ref. 17). These data sets provide an unprecedented opportunity to systematically map both 1D- and 3D-cooperation between DBPs. However, it is a great challenge to analyse this large amount of data and extract cooperation among multiple rather than pairs of DBPs.

To tackle this challenge and comprehensively catalogue DBP cooperation, we present here a new model to construct networks that represent both 1D- and 3D-association between DBPs. Analysing these networks in GM12878 and K562 has revealed complex cooperative relationships among TFs, histone modifications, chromatin-remodelling enzymes and chromatin architectural proteins. Through the identification of communities and cliques in the DBP cooperation network, we have uncovered many DBP interactions in the chromatin loop regions. Intriguingly, many of these 3D-cooperative DBPs directly interact with one another, which suggests their binding may be important for loop formation or stabilization in 3D space. Furthermore, we performed a comparative network analysis between GM12878 and K562, and revealed cell-type-specific cooperation between DBPs that are critical for regulating cell-type-specific functions.

## Results

**Gaussian graphical model.** To systematically identify DBP cooperation, we analysed DBP ChIP-seq data using Gaussian graphical model (GGM)[24]. GGM is an undirected probabilistic graphical model with the assumption that the data follows a multivariate Gaussian distribution with mean $\mu$ and covariance matrix $\Sigma$. Let $\Sigma^{-1}$ be the inverse of covariance matrix. If the $ij$th component of $\Sigma^{-1}$ is zero, then variables $i$ and $j$ are conditionally independent given all other variables in the network[24]. This important property serves as the foundation for GGM to infer direct interactions from high-dimensional data. Unlike relevance networks or correlation networks, in which edges are determined based on marginal correlations, GGM provides a stronger criterion of dependency, and thus further reduces the false positive rate. However, a great limitation of classic learning methods for GGM is the lack of sparsity in the resulting graph. A dense graph not only complicates downstream analysis but also raises the issue of overfitting the data. To cope with this, Friedman et al.[25] proposed an efficient algorithm, named graphical Lasso, to introduce sparsity to the GGM. Recently, Liu et al.[26] developed a data transformation method called Copula that can be used with the graphical Lasso algorithm to relax the normality assumption of GGM. Based on these recent advances, we developed a new framework to systematically identify cooperation between hundreds of DBPs.

Before applying the GGM to the DBP ChIP-seq data, we assessed its performance using synthetic data. First, we generated an Erdős–Rényi random graph as our ground truth (see Methods). To generate samples according to the simulated graph, we constructed a covariance matrix by assigning each $ij$th component a non-zero covariance if node $i$ and $j$ were connected in the simulated graph. All other components were then set to zero. We next drew samples from a multivariate Gaussian distribution parameterized by a zero mean vector and the constructed covariance matrix. These samples were used as input for network re-construction. As a comparison, we selected ARACNE[27], a popular algorithm for constructing gene regulatory networks that employs an information theory approach to infer interactions from gene expression data. We generated 10 networks with 50 nodes and another 10 networks with 100 nodes. When applying both methods to these data sets, we observed a superior performance of GGM with an average AUC of 0.923, which is significantly higher than ARACNE (AUC = 0.822) (Fig. 1a). This simulation showed that, when experimental data follows a Gaussian distribution, the GGM can precisely reconstruct the underlying graphical model. However, the real data can be quite noisy and may not be Gaussian distributed. To cope with this, we incorporated the Copula algorithm[26] and carried out a further benchmark to evaluate its performance on a more noisy data set. To produce synthetic gene expression data sets, we used GeneNetWeaver 3.1 (ref. 28), an in silico simulator that employs a dynamic model to simulate gene regulatory networks. The ground truth were subnetworks taken from yeast gene regulatory network with size 50 and 100, respectively. For each size, we performed 10 different simulations (network files and sample data are provided in the Supplementary Data 1). Again, GGM outperformed ARACNE (average AUC of 0.695 versus 0.615; Fig. 1b).

It is worth noting that GGM is much faster than ARACNE when the sample size is large. The time complexity for ARACNE is $O(N^3 + N^2M^2)$, where $N$ is the number of variables or nodes in the network, $M$ is the number of samples; as it scales with $M^2$, it is not suitable for our application where we have more than 10,000 samples (the number of ChIP-seq peaks). In contrast, GGM, with a time complexity $O(N^3 + N^2M)$, can easily handle a large number of samples. In practice, we observed that the GGM was 50–100 times faster than ARACNE on the synthetic data sets (Supplementary Table 1).

**Constructing the DBP cooperation network.** We applied the GGM framework to DBP ChIP-seq and Hi-C data, aiming to systematically detect DBP cooperation (Fig. 1c). We considered

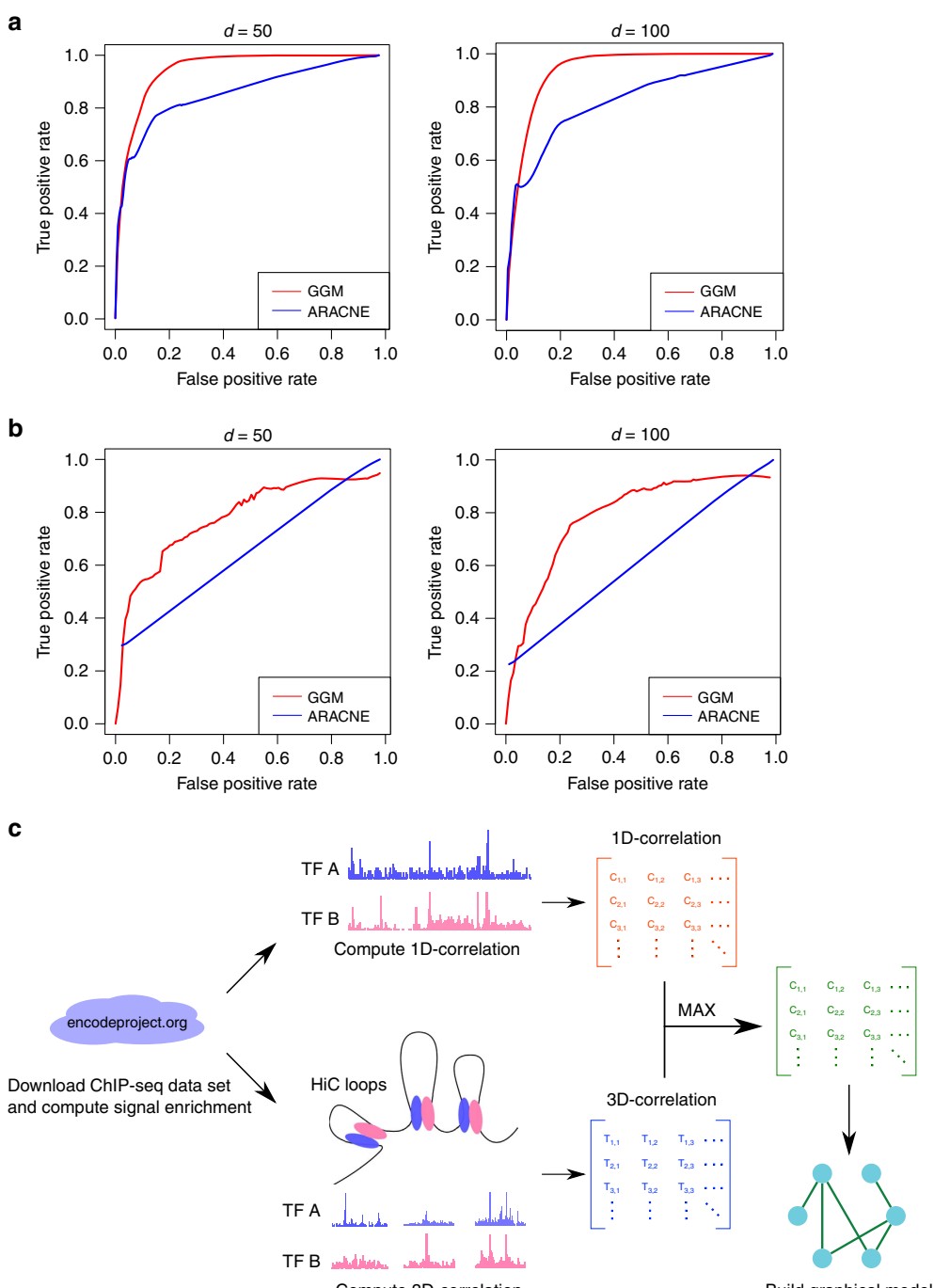

**Figure 1 | The performance of the GGM is consistently better than ARACNE.** Each plot shows the average curve from 10 independent simulations.
(**a**) ROC curve for samples generated from random networks. For each simulation 500 (left) or 1,000 (right) samples were generated from a network of 50 (left) or 100 (right) nodes. (**b**) ROC curve for samples generated from yeast sub-networks. For each simulation 500 (left) or 1,000 (right) samples were generated from a network of 50 (left) or 100 (right) nodes. (**c**) Workflow of the DBPnet pipeline.

both 1D (DBPs that bind to loci in the nearby linear genome) and 3D (DBPs that bind to loci that are spatially close but linearly distal in the genome) cooperation between DBPs. We first computed 1D and 3D correlation scores for each pair of DBPs separately using the 84 ChIP-seq data sets, including six histone modifications (H3K4me1, H3K4me3, H3K9me3, H3K27ac, H3K27me3 and H3K36me3) as well as chromatin loops called by the 5-kb resolution Hi-C data in a lymphoblastoid cell line GM12878 (ref. 17; see details in Methods). We then merged 1D and 3D correlation matrices by keeping the larger correlation

score at each entry. This matrix was used to construct the GGM, which represents the DBP cooperation network.

The DBP cooperation network (Fig. 2a) contains 484 associations between 84 DBPs. An edge between two proteins may indicate either a direct physical interaction or a co-occurrence of binding sites without direct interaction. To examine whether our model can recover protein–protein interactions (PPI), for each edge we searched for supporting evidence from the public PPI databases (Methods). Remarkably, 11% of edges (empirical $P$ value is 10e-9) in the GGM network

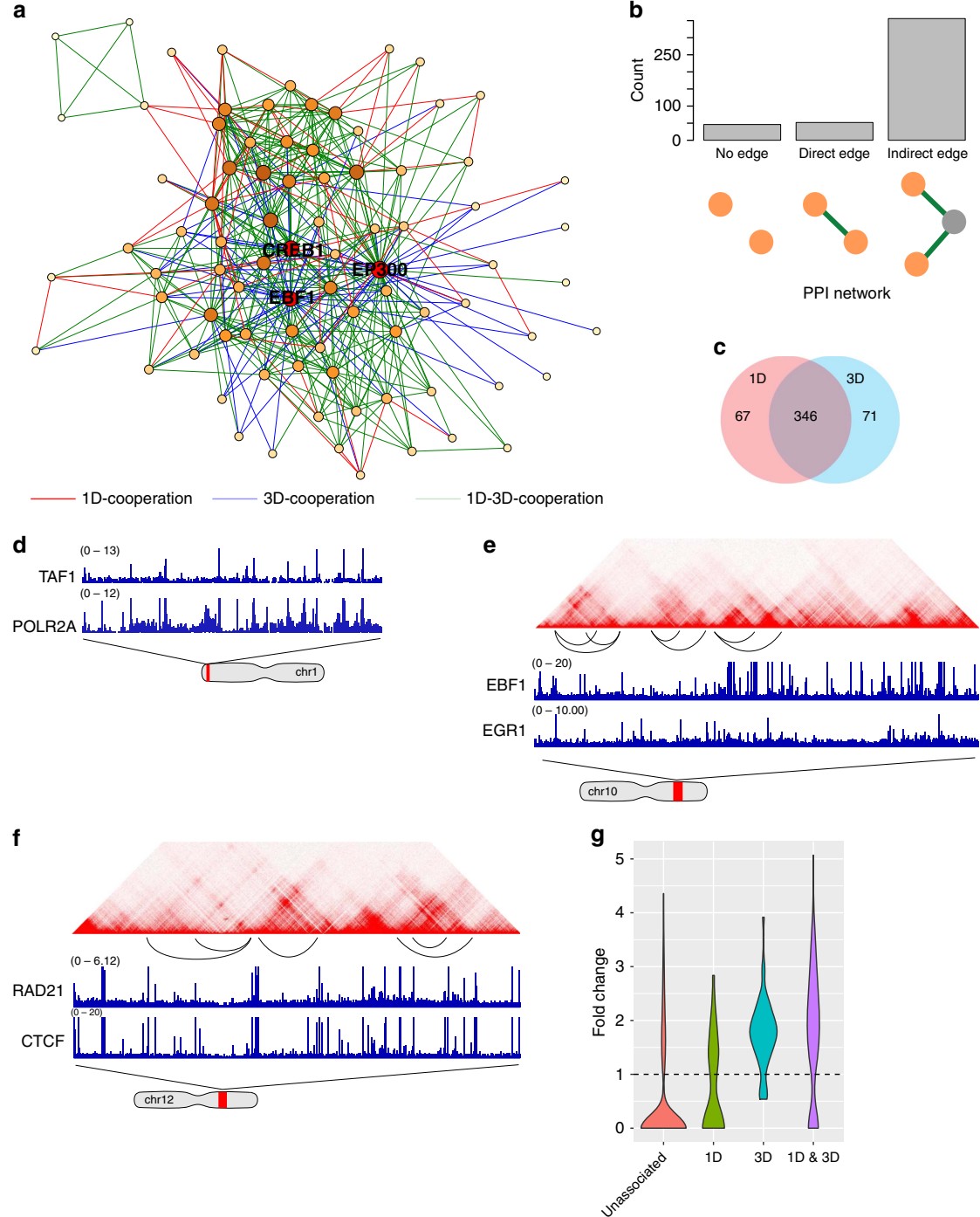

**Figure 2 | Constructing the DBP cooperation network in GM12878.** (**a**) DBP cooperation network in GM12878, with network hubs (EP300, EBF1, CREB1) being highlighted. (**b**) A significant portion of DBP cooperation is supported by evidence of direct protein–protein interactions. (**c**) The majority of DBP cooperation is a mixture of 1D and 3D cooperation. (**d**) An example of 1D-cooperation. (**e**) An example of 3D-cooperation. (**f**) An example of mixed cooperation. (**g**) Disease-associated genotype variations are enriched in 1D-dominant ($n = 67$), 3D-dominant ($n = 71$) and 1D–3D cooperative ($n = 346$) sites.

are also present in the PPI network (Fig. 2b). Another 80% of the associated DBPs are separated by one protein in the PPI network (the intermediate protein may not be analysed by the ChIP-seq experiments). This evidence strongly supports that the DBP cooperation recovered by our method is reliable and likely represents physical contacts. Furthermore, we found that 11.5% and 11.5% of 3D-dominant and 1D–3D cooperative edges, respectively, are coincident with protein–protein interactions, which is much higher than the 1D-dominant edges (1.8%);

94.6%, 63.5% and 79.4% of 1D-dominant, 3D-dominant and 1D–3D cooperative DBP pairs are separated by one protein in the protein–protein interaction network, respectively. This observation suggests that our analysis did identify physical interactions in the 3D space and many 1D-dominant ones may be formed through indirect interactions.

To characterize the topological properties of the DBP cooperation network, we plotted its node degree distribution. In agreement with other types of biological networks, we

observed that the node degree distribution of the DBP cooperation network follows a power law, reflecting its scale-free property (Supplementary Fig. 1). A prominent feature of scale-free networks is the existence of hubs, which are the highly connected nodes that may be critical for network stability. To identify hubs, we ranked the nodes in our network by two popular centrality metrics—node degree centrality and eigenvector centrality. Node degree centrality for a given node is simply the number of nodes that link to the given node, while eigenvector centrality reflects both the node degree and its connection with other well-connected nodes. We ranked the nodes by both their node degree and eigenvector centrality. The results show that EP300, CREB1 and EBF1 are the top three DBPs that have the best average rank (Fig. 2a and Supplementary Data 2). EP300 is an important cofactor that cooperates with many TFs[29,30] to perform a variety of biological functions. CREB1 plays a central role in the immune system through binding to the c-AMP response element, a ubiquitous DNA motif, to regulate gene transcription[31,32]. It was not surprising that these two general DBPs would be found as hubs. Previous studies showed that EBF1 is mainly expressed in B-lymphocytes (GM12878 is a lympoblastoid cell line) and is pivotal for maintenance of B-cell identity[33]. In the DBP cooperation network, EBF1 is linked to many important transcriptional regulators, including general activators such as EP300 and SP1, as well as B-lymphocyte-specific TFs such as PAX5, TCF12 and BCL11A (refs 34–37). By analysing the topology of the DBP cooperation network, we uncovered TFs that are crucial for cell functions.

**Identifying 1D and 3D cooperation between DBPs**. DBPs can cooperate through 1D or 3D interactions, which can be determined for each DBP pair using the constructed network. In this study, we define an edge as 1D or 3D cooperation if the 1D or 3D correlation score is larger than a pre-selected cutoff (0.3, see Methods). In GM12878, we found roughly the same numbers of 1D and 3D edges, 413 and 417 respectively. We noticed a great overlap between 1D and 3D edges (Fig. 2c). We thus labelled these DBP cooperations as 1D-dominant, 3D-dominant or 1D–3D cooperative (Supplementary Data 3).

A 1D-dominant association cooperation between two DBPs represents a frequent co-occurrence in linear space but not in the long-range interacting loci that form loops in the 3D space. In this category, we recovered some previously known interactions such as the RNA Pol II–TAF1 interaction[38] (Fig. 2d). Interestingly, we found 71 3D-dominant edges in GM12878. Most of these edges show weak 1D correlations but have significantly larger 3D correlations. For instance, the 1D and 3D correlation scores of EP300-MYC are 0.127 and 0.348 ($z$-score: −0.081 and 1.301), respectively. Indeed, a number of independent studies have shown that EP300 and MYC can cooperate to regulate gene transcription and the physical interaction between them has been previously reported[39,40]. We also identified novel 3D DBP cooperation. For example, EBF1 is an important TF in B lymphocytes, and Egr-1 is one of the key transcriptional regulators induced on B-cell antigen-receptor activation[41]. Both EBF1 and Egr-1 have crucial roles in B-cell development and differentiation. However, the interplay between these two proteins has not been reported. In our network, we found a 3D-dominant edge between EBF1 and Egr-1 (Fig. 2e), which suggests that they may form long-range loops to regulate cell-type-specific genes (4.4% of loop regions contain peaks of both EBF1 and Egr-1). Therefore, our framework provides a systematic way to uncover 3D cooperation between DBPs that are otherwise impossible to identify using previous approaches.

The 1D–3D cooperation is formed between DBP pairs with both 1D and 3D associations. Most associations fall into this category. A well-known example is CTCF-RAD21 (Fig. 2f). While 1D-dominant cooperation can be identified by previous approaches[19], the last two categories of DBP cooperation can only be identified through the integration of DBP ChIP-seq and Hi-C data, which highlights the advantage of our method.

To further confirm the importance of DBP cooperation, we analyzed the enrichment of genotype variations in regions bound by cooperative DBPs. The disease-associated genotype variations were downloaded from the NHGRI-EBI GWAS database[42]. Given two DBPs A and B, if they are 1D-cooperative, we considered sites bound by both A and B as foreground regions. If A and B are 3D-cooperative, we first identified chromatin loops where one of the two anchors is bound by A and the other is bound by B. Within these loops, we identified binding sites of A or B as the foreground. If A and B are 1D- and 3D-cooperative, we considered only those loops for which each of the two anchors contain the binding sites of both A and B, and these sites are used as the foreground. In all cases, background are the binding sites of A or B that are not in the foreground. We then calculated the percentage of regions containing genotype variations for foreground and background, and took the ratio as the genotype variation enrichment. In Fig. 2g, we showed that the vast majority of DBP cooperation are more enriched with disease-associated genotype variations. We performed the Mann–Whitney $U$-test to compare the significance level of enrichments of 1D-dominant, 3D-dominant and 1D–3D cooperative DBPs with that of uncooperative DBP pairs, the $P$ values are 1.6e-3, 5.2e-28 and 2.0e-45, respectively. These results suggest that DBP cooperation has important functional implications in a variety of diseases. Therefore, we anticipate that the binding sites of cooperative DBPs can be used to prioritize genotype variations to identify causal associations.

**Identifying DBP communities**. Modularity is an important property of biological networks. Characterization of the modularity in DBP cooperation networks can illuminate how multiple DBPs cooperate to carry out complex regulation. Modularity can be studied at different levels. For instance, cliques highlight local modules in the network while community structure is a more global view of the modularity. Communities are groups of nodes in a network that are more densely connected internally than with the rest of the network[43]. In other words, community structure is a partition or clustering of the nodes in a network. We applied the community detection algorithm[43] to the DBP network in GM12878, and the nodes are separated into four communities (Fig. 3a and Supplementary Table 2). From the community structure, an immediate observation is the existence of a very small community (yellow) that is formed by only five proteins, namely CTCF, RAD21, ZNF143, SMC3 and YY1. Intriguingly, all these five proteins are important in mediating chromatin looping[17,44,45], suggesting that a major function of this community may relate to chromatin structure organization. This observation suggests that DBPs in the same community are functionally cooperative, and the communities in the DBP network may have different biological functions.

To reveal the functions of a particular community, for each protein in the community we analysed its ChIP-seq peaks. We then ranked genomic loci by the number of proteins that bind to them and selected the top 5,000 loci as input to GREAT analysis to search for enriched GO terms. We found that the green community is linked to mRNA metabolic processes and translation-related functions (Supplementary Fig. 2a). The same analysis showed that the cyan community is also enriched with

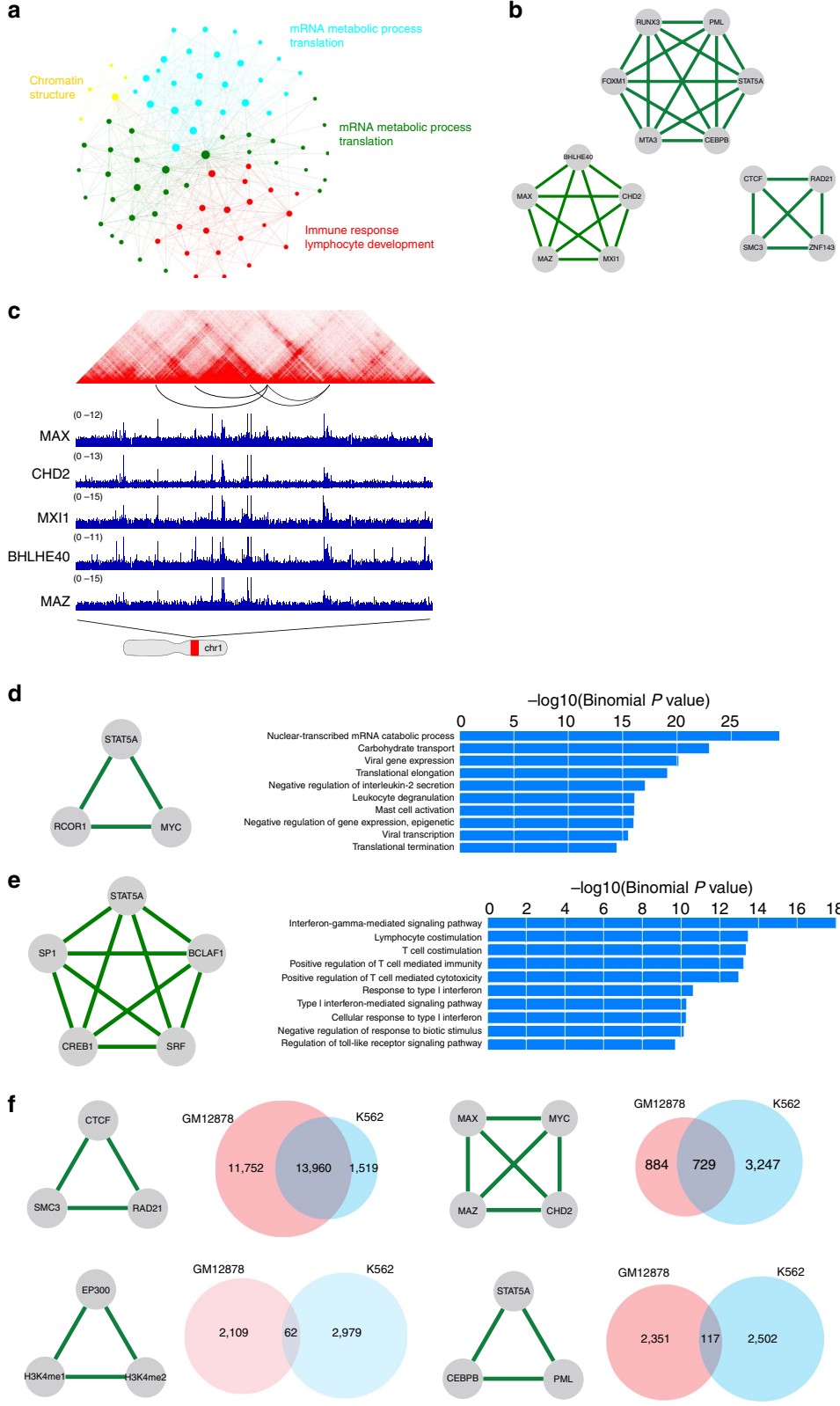

**Figure 3 | Network analysis reveals functions of DBP modules in GM12878 and K562.** (**a**) Communities in the DBP cooperation network and their functions. (**b**) Top DBP cliques. (**c**) An example of DBP cliques. (**d**) An example of K562-specific DBP cliques and the enriched GO terms of their binding sites. (**e**) An example of GM12878-specific DBP cliques and enriched GO terms of their binding sites. (**f**) Top conserved DBP modules in K562 and GM12878.

similar GO terms. Interestingly, these two communities share 3,248 out of the 5,000 loci used in GREAT analysis despite being segregated in the network (Supplementary Fig. 2b). A closer

examination of these two communities revealed distinct protein composition. For instance, all the six histone modifications as well as RNA polymerase II belong to the green community,

suggesting its pivotal role in gene transcription. In contrast, the cyan community contains numerous proteins, including ESRRA, BRCA1, NRF1, ETS1 and STAT3, that are involved in the oestrogen-signalling pathway.

Furthermore, GREAT analysis of binding sites of DBPs in the red community revealed that 'immune response', 'leukocyte activation' and 'lymphocyte activation' are the most enriched GO terms. These terms are highly specific to the B cell (Supplementary Fig. 2c), suggesting that this community is crucial in determining cell-type specificity. Indeed, many proteins in this community are known to be important for immune system development, such as STAT5A (ref. 46), BATF (ref. 47), BCL3 (ref. 48) and RELA (ref. 49).

**Identifying potential DBP complexes.** On a finer scale, the modularity of a network is revealed by cliques. A clique is a complete sub-graph in which every pair of nodes is connected. Intuitively, DBPs that form a clique in the network are more likely to function as a complex. Undoubtedly, the identification of such complexes is crucial for understanding the mechanisms of transcriptional regulation. Therefore, we searched for maximal cliques in the network and identified 220 cliques in GM12878 (Supplementary Data 4). We ranked DBP cliques by their average correlation scores for each DBP pair. We observed that edges in most of the top cliques are associated with high 1D and 3D correlation scores, which suggests that they are likely to form complexes mediating chromatin loop formation. Figure 3b shows the top three highest ranked cliques. Next, we checked the percentage of shared peaks in the union of all DBP peaks and identified the loops that overlap with these shared peaks. As a comparison, for each k-component DBP clique, we randomly selected k DBPs and did the same analysis. We took 50,000 samples and used them as a null model for enrichment and empirical P value calculation. As a result, all the DBPs in the cliques share a significant amount of peaks that occur in loops (Table 1), which confirmed the co-occurring bindings of the DBPs in a clique.

The top-ranked clique is CTCF-RAD21-SMC3-ZNF143. RAD21 and SMC3 are components of the cohesin complex. Cohesin is a multi-subunit protein complex and plays an essential role in sister chromatid cohesion and chromosome segregation during cell division[50]. Cohesin is also crucial for regulating gene expression and mediating chromatin long-range interactions[51]. Cohesin-dependent chromatin interactions are usually mediated by the cooperation of cohesin and CTCF[45]. The involvement of ZNF143 in this complex has also been reported[44]. ZNF143 is believed to provide sequence specificity for chromatin interactions[52]. Overall, our analysis successfully recovered this important and well-characterized loop-forming complex.

The other two cliques, PML-FOXM1-MTA3-STAT5A-CEBPB-RUNX3 and MAX-MAZ-MXI1-CHD2-BHLHE40 (Fig. 3c) have not been reported. STAT5A and RUNX3 are two of the major

transcription factors that play essential roles in lymphocyte development. The physical interaction between STAT5 and RUNX3 has been reported[53]. Moreover, CEBPB binds to RUNX2 that has been shown to be associated with RUNX3 (refs 54,55). To investigate the function of this module, we extracted all loci bound by these six DBPs and performed GREAT analysis. The most significant GO terms are 'immune response', 'leukocyte activation' and 'lymphocyte activation'. These results suggest that this module may play important roles in the development of lymphocytes.

The functions of the MAX-MAZ-MXI1-CHD2-BHLHE40 clique are more general. The enriched GO terms for their binding sites are 'ribonucleoprotein complex biogenesis', 'nuclear-transcribed mRNA catabolic process ribosome biogenesis' and 'translation'. The interaction between MAX and MXI1 is well studied[56] but interactions between other proteins have not been reported. However, the functions of these proteins are highly related. For example, BHLHE40 is a repressor that can interact with and recruit HDACs, which suggests a role for BHLHE40 in chromatin remodelling. CHD2 is also a chromatin remodeler. These observations suggest that DBPs in this clique may act together to alter chromatin states and regulate gene translation.

**Comparative analysis of DBP cooperation networks.** DBPs have different cooperative modes in different cells. To perform a comparative analysis of the networks in different cell types, we focused on 68 proteins for which ChIP-seq data sets are available in both K562 and GM12878, and constructed TF cooperation networks in these two cell types.

To find cell-type-specific DBP cliques, we first identified cell-type-specific edges in the 68-node networks. We then searched for cliques in both GM12878- and K562-specific networks that consist of edges present in one but not the other cell line. We found 74 and 7 cell-type-specific cliques for GM12878 and K562, respectively (Supplementary Data 5). Cell-type-specific cliques shed light on how cells achieve transcriptional specificity through the combinatorial regulation of DBPs. For example, STAT5A is a member of STAT protein family. It is activated by a number of cytokines and plays a central role in the development of many different organs. However, how STAT5A cooperates with other DBPs to carry out cell-type-specific regulation is largely unknown. Our analysis showed that STAT5A, together with MYC and RCOR1, forms a clique in K562, which is absent in GM12878. MYC is an oncogene and has been shown to play a critical role in leukaemia formation[57,58]. STAT5A-MYC cooperation may be important to maintain the state of leukaemic cells. To further characterize the functions of the STAT5A-MYC-RCOR1 clique, we performed GREAT analysis on loci bound by all the three proteins in K562 and identified functions specific to leukocyte, such as 'leukocyte degranulation', 'regulation of interleukin-2 secretion' and 'mast

**Table 1 | The top 3 most frequently occuring DBP cliques.**

| DBP clique | No. of sites/percentage/enrichment (P value) | No. of overlapped loops/percentage/enrichment (P value) |
|---|---|---|
| CTCF–RAD21–SMC3–ZNF143 | 13,829/21.9%/8.8 (4.0e-5) | 5,668/47.2%/30.5 (<2.0e-5) |
| PML–FOXM1–MTA3–STAT5A–CEBPB–RUNX3 | 2,195/3.1%/16.8 (1.8e-3) | 388/3.2%/16.3 (5.2e-4) |
| MAX–MAZ–MXI1–CHD2–BHLHE40 | 2,900/8.8%/24 (8.0e-5) | 501/4.2%/12.5 (1.1e-3) |

The central column gives the number of regions bound by all DBP members in a clique, the percentage of regions bound by all DBP members, their fold enrichment over background and the empirical P values. The right column gives the number of loops that are overlapped with the DBP-binding sites, the percentage, the fold enrichment over background and empirical P values.

cell activation' (Fig. 3d). These functions are drastically different from those enriched in GM12878 where STAT5A is associated with BCLAF1, SRF, CREB1 and SP1; GREAT analysis on the shared peaks suggests this clique is involved in lymphocyte specific functions (Fig. 3e).

Next, we sought to identify common DBP cliques in GM12878 and K562. First, we extracted a common network using edges shared by the two networks. We then searched for cliques in this network. We identified CTCF–RAD21–SMC3, MAX–MYC–MAZ–CHD2, EP300–H3K4me1–H3K4me2 and STAT5A–CEBPB–PML as top-ranked cliques (Fig. 3f). CTCF–RAD21–SMC3 interaction is known to be conserved across different cell types and it is not surprising that this clique is shared between the two cells. In the MAX–MYC–MAZ–CHD2 clique, MAX–MYC–MAZ is also a well-known complex that is found in multiple cell lines but their interaction with CHD2 has not been reported. The involvement of the chromatin-remodelling gene CHD2 in the MAX–MYC–MAZ complex suggests MAX–MYC–MAZ may utilize CHD2 to modify chromatin structure and alter gene expression. EP300–H3K4m1–H3K4me2 represents an enhancer's signature, and has been found in many cell types. In the clique of STAT5A–CEBPB–PML, there is evidence for the STAT5A–CEBPB interaction: STAT5A was demonstrated to cooperate with CEBPB to regulate gene transcription[59]; STAT5A can induce deacetylation of CEBPB[60]. Their interaction with PML is less well-studied but STAT5 is shown to be activated by the PML/RARα fusion protein in acute myeloid leukaemia[61]. These common cliques in both GM12878 and K562 indicate their cell-type-independent cooperation.

We next investigated whether these common cliques bind to the same loci in the two cells. For each clique, we identified the sites bound by all the member DBPs and counted how many of them are shared between the two cell types. We observed that the CTCF–RAD21–SMC3 clique shared 13,960 (51.3%) common binding sites in K562 and GM12878 (Fig. 3f), which is in agreement with the general roles of CTCF and the cohesin complex in stabilizing loops[17]. The MAX–MYC–MAZ–CHD2 clique shows moderate conservation with 729 (15%) common binding sites across the two cell types. In contrast, the binding sites of P300–H3K4me1–H3K4me2 and STAT5A–CEBPB–PML are highly cell-type-specific. Since P300–H3K4me1–H3K4me2 mark active enhancers and enhancers are highly cell-type-specific, it is understandable that there are only 62 (1.2%) P300–H3K4me1–H3K4me2 peaks shared across cell types. The fact that only a small percentage (2.4%, 117 sites) of STAT5A–CEBPB–PML sites are shared between GM12878 and K562 is surprising. To investigate the reason why STAT5A–CEBPB–PML has distinct binding profiles in the two cell types, we first analysed the enriched GO terms for the sites bound by all three DBPs in K562 and GM12878, respectively. Enriched GO terms in each cell type are highly specific: the top terms in GM12878 are 'immune response' and 'lymphocyte activation'; sites in K562 are enriched with GO terms such as 'platelet activation' and 'erythrocyte differentiation', which are highly specific to K562.

The above analyses show that the same DBPs can bind to different loci to regulate cell-type-specific functions. There are several possible reasons for such cell-type-specific binding, such as differential accessibility of chromatin, DBPs recognizing cell-type-specific motifs[62], and DBPs partnering with different cofactors. To understand the differential binding of the STAT5A–CEBPB–PML clique, we identified their cell-specific partners by examining the binding peaks of all the available ChIP-seq data in the regions bound by STAT5A–CEBPB–PML in GM12878 and K562 (Fig. 4a,b). It is obvious that the co-occurring DBPs are very different in the two cell types:

RUNX3, BCL11A, BATF in GM12878 compared with TEAD4, TAL1, GATA2 in K562. To assess the contribution of cofactors to such cell-type-specific binding, for each potential cofactor we used its ChIP-seq peaks to discriminate binding regions of STAT5A–CEBPB–PML in GM12878 and K562. The top 12 TFs that have best discriminative accuracy are RUNX3, TEAD4, TAL1, BCL11A, BATF, IRF4, PAX5, POU2F2, BCL3, GATA2, MYC and EBF1 (Supplementary Data 6). Strikingly, either RUNX3 or TEAD4 alone can achieve an over 99% accuracy, which is consistent with their distinct binding patterns in Fig. 4. When the binding sites of these 12 TFs were used together to train a logistic regression model, we achieved a 100% accuracy rate for discriminating the STAT5A–CEBPB–PML binding regions in the two cell lines. Therefore, the cell-type-specific binding of this DBP clique can be explained by its partnership with different cofactors. Furthermore, we observed that all 12 TFs except MYC are differentially expressed in the two cell types (Supplementary Data 6), suggesting that the cell-type-specific binding of this DBP clique is largely due to cell-type-specific expression of cofactors. Because of the limited number of ChIP-seq experiments, possible partners might not be profiled. Therefore, we performed de novo motif analysis using MEME–ChIP[63] in STAT5A–CEBPB–PML sites and then matched the found motifs to the known ones. Clearly, the de novo motifs found in K562 and GM12878 were very different. Encouragingly, the motifs of several co-factors identified from ChIP-seq experiments were also retrieved from the de novo motif analysis. These results suggest that the STAT5A–CEBPB–PML complex indeed has different regulatory mechanisms in different cell types. To further interrogate the regulatory mechanisms of their cooperation, we used Spamo[64] to find spacing constraints between de novo motifs. As a result, in GM12878 we found two de novo motifs, corresponding to STAT5A and MEF2, showing a statistically significant spacing constraint with the MEF2 motif occurring 13 bp downstream of the STAT5 motif (Fig. 4c). This finding is new as there is no previous report about the partnership of STAT5A and MEF2. We also found, in K562 the TAL1::GATA1 motif frequently appears upstream of RUNX1 sites at a distance of 38 bp. GATA and RUNX usually cooperate with each other and form a cis-regulatory module[65,66]. Therefore, our analysis has identified both new and known spacing constraints between TFs.

## Discussion

We present here a first systematic search of DBP complexes mediating chromatin loop formation using a novel framework. Our method can identify both 1D and 3D cooperation between DBPs. Many of the identified cooperations are likely a result of physical interactions as most of the edges in the DBP cooperation network are supported by the PPI data. Our results showed that 3D-cooperation between TFs is ubiquitous, indicated by 86% of identified associations having strong 3D correlation scores, which can only be discovered by integrating DBP binding and chromatin structure data. The 3D-cooperation most often accompanies 1D-cooperation as the majority (71%) of DBP cooperation is a mixture of 3D and 1D events. Furthermore, we observed enrichment of disease-associated genotype variations in DBP cooperative binding sites, which suggests the functional importance of DBP cooperation.

Identification of cooperation between multiple DBPs has been a challenging problem. Combinatorial approaches are limited to consider cooperation between a small number of DBPs because of the exponential increase of the possible combinations. In contrast, our model can easily search combinatorial cooperation in thousands of DBPs. By identifying modules and cliques in the network, we have uncovered closely collaborated DBPs,

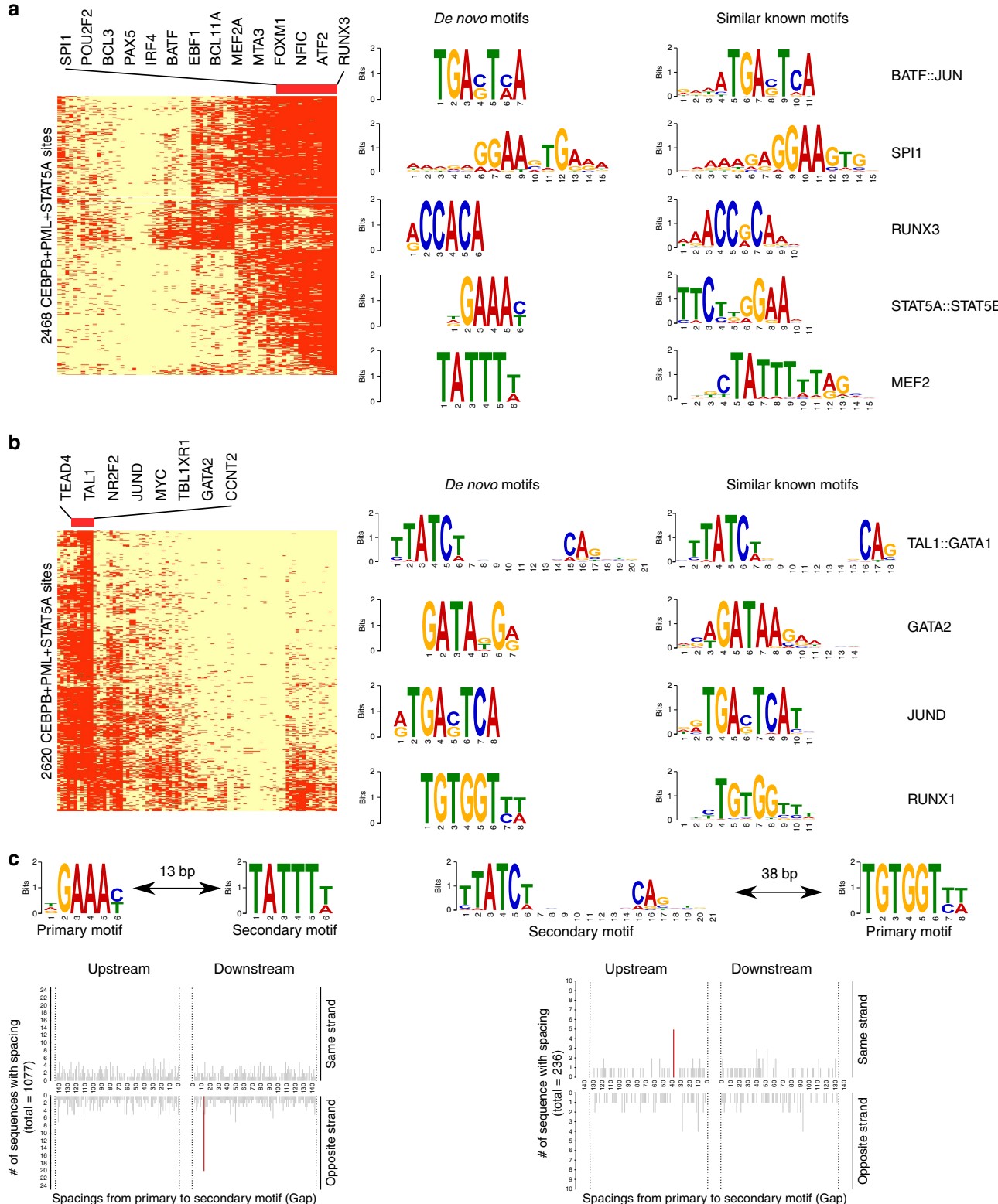

**Figure 4 | CEBPB–PML–STAT5A cooperates with different DBPs in GM12878 and K562.** (**a**) DBP-binding profile (left) and enriched *de novo* motifs (right) in 2468 CEBPB–PML–STAT5A-binding sites in GM12878. (**b**) DBP-binding profile (left) and enriched *de novo* motifs (right) in 2620 CEBPB–PML–STAT5A-binding sites in K562. (**c**) Enriched spacing between *de novo* motifs found in CEBPB–PML–STAT5A sites in K562 and GM12878.

particularly those associated through 3D interactions in chromatin loops that may be crucial for loop formation or stabilization.

Our comparative analyses between GM12878 and K562 reveals different mechanisms of achieving cell-type specificity: using different combinations of DBPs or using the same protein complex but collaborating with different partners. Interestingly, we also found spacing constraints between the binding sites of certain partners, which implies higher-order regulatory rules for not only 1D but also 3D DBP cooperation. One major limitation of this work is that it relies on high resolution Hi-C data that is

only available in a limited number of cell types. Furthermore, the chromatin loops used in this project are taken directly from the study of Rao et al.[17] that were defined using very conservative criteria. Additional loops might be identified using less conservative criteria or other technologies. However, as the sequencing technology rapidly evolves, these limitations will be overcome by the availability of more and more ChIP-seq and Hi-C data at even higher resolution. In conclusion, our model provides a powerful tool for integrative analysis of DBP binding and chromatin structure data in different cell types, which will facilitate the uncovering of the molecular mechanisms for transcriptional regulation and 3D chromosome organization.

## Methods

**Data sets.** BAM files of ChIP-seq experiments in K562 and GM12878 were downloaded from the ENCODE project website[67]. Chromatin loops were downloaded from the study by Rao et al.[17]. Because only these two cell lines had both DBP ChIP-seq and 5-kb resolution Hi-C data, we focused on these data sets in this study.

**Data preprocessing.** We divided each chromosome into consecutive 1-kb regions. For each protein, we computed the Reads Per Kilobase per Million (RPKM) mapped reads on these regions. The fold enrichment was calculated using MACS's algorithm[68] with customized parameters. In particular, we set $\lambda_{local} = \max\{\lambda_{BG}, \lambda_{14k}, \lambda_{24k}\}$ where $\lambda_{BG}$ is the average RPKM of the whole genome; $\lambda_{14k}$ and $\lambda_{24k}$ are average RPKM of 14 and 24 kb windows. We used a larger window size than MACS's default size and a loose $P$ value (0.01) to call peaks to increase the sensitivity for detecting broad peaks. For each 1 kb region, if it was called as a peak, we used the fold enrichment as its ChIP-seq enrichment score; otherwise, a zero score is assigned to that region. After computing the enrichment scores for every protein, we removed regions with low variation of scores by requiring the s.d. of scores to be at least 1. This excludes some unwanted artefacts from our analysis. For instance, regions with low mappability or an abnormally high signal[67]. Next, for each DBP pair we calculated the Spearman's correlation of ChIP-seq enrichment scores in the remaining bins as the 1D-correlation score. To compute 3D correlation scores, we first downloaded the 3D interaction loops identified in a 5-kb resolution Hi-C study[17]. Next, for each DBP we computed its enrichment scores on loop regions as follows: suppose we have $n$ loops, denoted by $L^1, L^2,\ldots, L^n$ and each loop $L^i$ consists of two interacting loci $L_a^i$ and $L_b^i$. To compute the enrichment score of a given DBP on loop $L^i$, we first binned $L_a^i$ into 1-kb regions, and then took the maximum of ChIP-seq enrichment scores of these bins as the enrichment score for $L_a^i$. Likewise, we can compute the enrichment score for $L_b^i$. Then, given a pair of DBPs denoted by $A$ and $B$, for every loop, we first compared the enrichment scores of $A$ on the two interacting loci. We considered the interacting locus with larger enrichment score of protein A as A's primary binding locus, and the other interacting locus as the primary binding locus of protein B. The enrichment scores of primary binding loci for each protein were then used to compute the correlation coefficient.

**Network construction.** We adapted the GGM to construct the DBP cooperation networks. GGM assumes that the observations have a multivariate Gaussian distribution with mean $\mu$ and covariance matrix $\Sigma$. Let $\Sigma^{-1}$ be the inverse of covariance matrix. If the $ij$th component of $\Sigma^{-1}$ is zero, then variables $i$ and $j$ are conditionally independent given other variables. Therefore, each non-zero component represents an edge in the network. To efficiently and accurately estimate the inverse of the covariance matrix using DBP ChIP-seq data, we employed the Graphical lasso algorithm[25] and the Copula method[26]. We used a lasso penalty equal to 0.3 in this study. We chose this value because <15% of DBP pairs have a correlation score >0.3. To estimate the false discovery rate, we generated a null model by random shuffling of DBP binding sites to represent uncooperative DBPs. When we applied our algorithm to this data set, the cutoff we chose identifies zero cooperation, suggesting our method has a very low false discovery rate. Because we aimed at identifying DBP interactions, edges with negative correlations were removed in the network analysis.

**Network analysis.** We used Eppstein's algorithm[69] for maximal clique searching, which gives an exact solution in near optimal time. For community detection, we used Newman's algorithm[43].

**Comparing the DBP cooperation network with PPI network.** Protein-protein interaction data was obtained from the BioGrid database[70] version 3.2.99. For each edge formed by node A and B in a DBP cooperation network, if also present in the PPI network, it was considered as a direct interaction. Otherwise, we checked whether there exists a third node in the PPI network that connects to both A and B; if so, this edge was considered as an indirect interaction. To determine the statistical significance of these overlaps, we first replaced the nodes in the DBP cooperation network with randomly selected genes from the PPI network. Next, we counted the direct and indirect interactions in the simulated network. This process

was repeated for $10^9$ times to generate the background distribution, which was then used to calculate the $P$ values.

**Simulated networks.** To generate an Erdős–Rényi random graph, we used the $G(n, p)$ model. This model specifies an $n$-node network, in which each edge is included with a probability $P$ independent from every other edge. We used $P = 0.2$ in this paper, which gives rise to a sparse network. We follow the procedures given in ref. 26 to generate a Gaussian distributed data set that was used for constructing the simulated network.

We used Genetweaver 3.1 to extract random subnetworks with different sizes (50 and 100 nodes) from the yeast gene regulatory network provided by the software. For other parameters, we used the software's default setting.

To draw the receiver operating characteristic (ROC) curve, we counted the number of true positives, false positives, true negatives and false negatives. If a predicted edge is present in the true network, it is a true positive, otherwise it is a false positive. Edges that are present in the true network but not identified by the algorithm are defined as false negatives. True negatives are edges that are not present in either predicted or true networks.

**Code availability.** The code used in this study is available at http://wangla-b.ucsd.edu/star/DBPnet/index.html.

**Data availability.** Chromosome loops in GM12878 and K562 were available in Gene Expression Omnibus (GEO) repository with the accession code GSE63525. ChIP-seq data sets analysed in this study were available from https://www.enco-deproject.org with accession codes listed in Supplementary Data 7.

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

## Acknowledgements

This work was partially supported by NIH (U54HG006997) and CIRM (RB5-07236). K.Z. was partially supported by UCSD Frontiers of Innovation Scholars Program and Dr Huang memorial scholarship.

## Author contributions

K.Z. and W.W. developed the method and analysed the results. N.L. performed the PPI network analysis. W.W. supervised the entire project. K.Z. and W.W. wrote the manuscript with contribution from R.I.A.

## Additional information

**Competing financial interests:** The authors declare no competing financial interests.

