## [Peer Review File · Nature Communications]

Reviewers' comments:

Reviewer #1 (Remarks to the Author):

In this project, the authors devised a novel algorithm to identify the combinatorial effects of transcription factors by integrating both the 2D epigenomics profiling data and 3D chromatin interaction data. The approach is refreshing and the results are convincing.

Most of the previous literatures on similar subject have been focused on TF interactions or the combinatorial effect in 2D space, when two TF binding sites are located next to each other. This method will work to a certain degree, but will definitely miss some important information as 3D genome organization have shown to be strongly related with gene regulation.

Instead, Wang group built a Gaussian Graphical model and used multiple sources of TF interaction data, including 2D proximity, 3D proximity, and Protein-protein interaction. Interestingly, they recovered 71 3D-dominant edges in GM12878 with weak 2D correlation. More importantly, they found CTCF-RAD21-SMC3 clique are shared between different cell types, but enhancer associated clique such as STAT5A-CEBPB-PML and P300-H3K4me1-H3K4me2 are more tissue-specific. Overall, this is novel and exciting work. The conclusions are well supported by evidence and I recommend publication.

Some minor suggestions that can help improve the paper:

1. Can authors comment why they called peaks themselves rather than directly used the peaks provided by the ENCODE consortium?
2. Fig. 1C, can authors describe how much difference is there between the 2D correlation matrix and 3D matrix? For example, what percentage of the data entries doesn't agree between the two tables.
3. "... gene expression depend on local chromatin states and three-dimensional (3D) organization[1-9]." Please cite Dixon et al Nature 2012, Shen et al Nature 2012.

Reviewer #2 (Remarks to the Author):

The authors report a new algorithm combining ChIP-seq and high-resolution Hi-C datasets to infer cooperative (and presumably regulatory) networks of chromatin factors, both at the "2D" level (factors sharing genomic binding sites) and the "3D" level (factors participating at the anchor sites of chromatin loops). This analysis is to my knowledge the first of its kind to fully integrate both types of information; while it relies on correlations and presents no functional experiments to demonstrate causal relationships, the approach appears robust and should stimulate future work and discussions, so I support the manuscript's publication. There are a few points which, if addressed, would strengthen the manuscript. However, the feasibility of such work is unclear, so may be suitably addressed by just mentioning them in the discussion.

1. Is there anything particularly special about the factors forming then nodes within the DBP cooperation network? Could these not be simply explained as corresponding nodes in protein-protein interaction networks (i.e. the factors that form the greatest repertoires of protein-protein interactions necessarily cooperate with more other factors, both at 2D and 3D level)?
2. If, as both the authors and the literature suggest, chromatin looping is largely linked to protein-protein interactions, it is reassuring that the majority of cooperators do so in both 2D and 3D. However, it is conceptually difficult to explain 3D DBP cooperation which does not have a corresponding 2D one. Could these just be a consequence of a "six degrees of separation", whereby

both partners of a 3D-exclusive cooperation (A and B) are actually in a "true" cooperation with C and D respectively, and the fact that C and D themselves cooperate brings along A and B indirectly? Is there a way to decouple these effects in the analysis?

3. Similar to point 2, is there a way to test whether 2D AND 3D cooperation links to genotype variations are synergistic, compared to 2D only or 3D only?
4. Is there a technical or biological explanation for the separation of the green and cyan communities? Do factors in one or other community "actively segregate"?
5. The supposed "rewiring" of DBP cliques across cell types is very interesting. Although exhaustive ChIP-seq datasets are not available to fully address this, is there a way to estimate how much of this rewiring can be accounted for by cell-specific factors participating in the complexes? If it's not 100%, what other factors could be responsible - a differential accessibility to cognate binding sites?

Other small modifications to consider:

1. On the first page, "mediator" is a poor choice of word, as it could be confused with components of the Mediator complex, which also participate in looping interactions.
2. Although I sort of worked it out, a better and more formal explanation of the difference between a "community" and a "clique" would make the manuscript much easier to follow.
3. The manuscript is easy enough to follow, but should be proof-read by a native English speaker, as there are many small grammatical mistakes throughout.
4. It should be pointed out that a major limitation of this work is that it relies on very high-resolution Hi-C data, which is able to identify peaks over a sufficiently small window size. So far, this has only been achieved by the VERY EXPENSIVE work of Rao et al. (cited in the manuscript). Presumably, more datasets will follow, but it is a current bottleneck for the wider application of this work.

Reviewer #3 (Remarks to the Author):

NCOMMS-16-04035 Review

Zhang, Li & Wang: Systematic identification of protein combinations mediating chromatin looping

The authors perform a meta-analysis of 2D and 3D colocalization data to identify cooperating complexes that mediate 3D loop assembly. The idea is creative and the bioinformatics thought-provoking. Identification of combinatorial complexes that specify 3D loop formation would advance the field. Unfortunately, the authors claim correlative associations but fail either to demonstrate sufficient statistical rigor, or to test their claim that cooperation between the identified proteins indeed regulates 3D architecture. The authors should either more clearly and convincingly underwrite their bioinformatic meta-analysis, or carry out functional tests, or both.

Major concerns

Points 1-3 below concern the rigor of the bioinformatics, particularly the determinations of false discovery rates and the strength of the correlations between protein occupancy and loop assembly genome-wide:

1. The ROC curves presented in Fig. 1 only address simulations. The authors do not present an estimate of FDR using real data, so the reader cannot assess the validity of their analysis method.
2. In Fig. 3B, the "enrichment" value is not translated into a metric that allows evaluation of the statistical significance of a given clique. The authors should add to the table some measure of significance, the operational equivalent of a p-value.

3. The strength of the correlation between 2D colocalization and 3D association for the bulk population of identified sites is not clear. Although the chosen examples in Figures 2E, 2F, and 3D are convincing, a global visualization that displays all identified sites would help convince the reader that the thousands or so sites are indeed enriched.

Points 4-7 below concern the functional significance of the correlations observed here:

4. The authors define protein "communities" based on network analysis and assign functions based on GO ontology terms. It seems likely that these assemblies include many enhancer-promoter interactions that may regulate the genes in these processes. Is it possible to build on a functional correlation by identifying contacts with promoters identifying enriched GO ontology?

5. Two new potential structural complexes are inferred, PML-FOXM1-etc. and MAX-MAZ-etc. Functional tests of these presumptive complexes are needed. Does knockdown of one component affect 2D binding of another? How many components are necessary to maintain the 3D loop?

6. The inferred cell type-specific complexes, such as those containing STAT5, require functional tests. Such experiments could present a wonderful opportunity to demonstrate that combinatorial control imparts context-specific 3D function.

7. The assertion that the GGM network discovers direct interactions overreaches, because protein-protein interaction databases include findings from procedures that detect indirect associations, e.g., immunoprecipitation followed by mass spec. Follow up studies, such as in vitro binary interaction data, and mutations that abrogate the interaction in vitro and in vivo, would be needed to conclude direct protein-protein interaction. The authors should to soften their claims about the directness of the inferred interactions.

REVIEWERS' COMMENTS:

Reviewer #1 (Remarks to the Author):

The authors have addressed all my concerns. I recommend the publication of this work.

Reviewer #2 (Remarks to the Author):

The authors have satisfactorily replied to all comments, and I support the manuscript for publication.

Reviewer #3 (Remarks to the Author):

The authors have successfully extracted new information from the large data sets they analyzed, and addressed thoughtfully most of my concerns. I am comfortable with acceptance of this revised version.

We are very grateful to the constructive comments from the three reviewers and have revised the manuscript accordingly. We have changed "2D cooperation" to "1D cooperation" through the text to better reflect the two proteins' binding to the linear genome. But in the response letter we still use 2D cooperation for the sake of discussion convenience. The following is a point-to-point response.

Reviewer #1 (Remarks to the Author):

In this project, the authors devised a novel algorithm to identify the combinatorial effects of transcription factors by integrating both the 2D epigenomics profiling data and 3D chromatin interaction data. The approach is refreshing and the results are convincing.

Most of the previous literatures on similar subject have been focused on TF interactions or the combinatorial effect in 2D space, when two TF binding sites are located next to each other. This method will work to a certain degree, but will definitely miss some important information as 3D genome organization have shown to be strongly related with gene regulation.

Instead, Wang group built a Gaussian Graphical model and used multiple sources of TF interaction data, including 2D proximity, 3D proximity, and Protein-protein interaction. Interestingly, they recovered 71 3D-dominant edges in GM12878 with weak 2D correlation. More importantly, they found CTCF-RAD21-SMC3 clique are shared between different cell types, but enhancer associated clique such as STAT5A-CEBPB-PML and P300-H3K4me1-H3K4me2 are more tissue-specific.

Overall, this is novel and exciting work. The conclusions are well supported by evidence and I recommend publication.

Response: We thank the reviewer for the positive comments.

Some minor suggestions that can help improve the paper:

1. Can authors comment why they called peaks themselves rather than directly used the peaks provided by the ENCODE consortium?

Response: We did not use the peaks provided by the ENCODE consortium because we want to reduce false negatives and do not miss any possible DBP binding sites. We thus used a loose cutoff in peak calling because false positives are not an issue in our application. Moreover, a loose cutoff works best for DBPs with broad peaks like histones. On the contrary, the Irreproducible Discovery Rate (IDR) framework used by ENCODE identified only the most confidential peaks and do not work well for DBPs with broad peaks and the ENCODE has only provided peaks for mainly transcription factors that have very sharp peaks. A brief explanation was included in the text (page 10 in Methods).

2. Fig. 1C, can authors describe how much difference is there between the 2D correlation matrix and 3D matrix? For example, what percentage of the data entries doesn't agree between the two tables.

Response: The values in the 2D- and 3D- correlation matrices are quite different. However, 346 pairs of DBPs showed high correlations in both 2D- and 3D- correlation matrix, which accounts for 71.4% of the total associations (Fig 2).

3. " ... gene expression depend on local chromatin states and three-dimensional (3D) organization[1-9]." Please cite Dixon et al Nature 2012, Shen et al Nature 2012.

Response: We thank the reviewer for pointing out the missing references and they were added in the revised manuscript.

Reviewer #2 (Remarks to the Author):

The authors report a new algorithm combining ChIP-seq and high-resolution Hi-C datasets to infer cooperative (and presumably regulatory) networks of chromatin factors, both at the "2D" level (factors sharing genomic binding sites) and the "3D" level (factors participating at the anchor sites of chromatin loops). This analysis is to my knowledge the first of its kind to fully integrate both types of information; while it relies on correlations and presents no functional experiments to demonstrate causal relationships, the approach appears robust and should stimulate future work and discussions, so I support the manuscript's publication. There are a few points which, if addressed, would strengthen the manuscript. However, the feasibility of such work is unclear, so may be suitably addressed by just mentioning them in the discussion.

Response: We thank the reviewer for the positive comments.

1. Is there anything particularly special about the factors forming the nodes within the DBP cooperation network? Could these not be simply explained as corresponding nodes in protein-protein interaction networks (i.e. the factors that form the greatest repertoires of protein-protein interactions necessarily cooperate with more other factors, both at 2D and 3D level)?

Response: This is a great point. Indeed, we showed that a significant portion of associated factors do form protein-protein interactions: 11% of edges (p-value is $10e-9$) in the GGM network are also present in the PPI network (Fig. 2b) and another 80% of the associated DBPs are separated by one protein in the PPI network (the intermediate protein may not be analyzed by ChIP-seq experiments). This concordance between our analysis and PPI network suggests that our method can detect factors mediating chromatin looping through forming protein complexes that bring distal loci to close spatial proximity.

2. If, as both the authors and the literature suggest, chromatin looping is largely linked to protein-protein interactions, it is reassuring that the majority of cooperators do so in both 2D and 3D. However, it is conceptually difficult to explain 3D DBP cooperation which does not have a corresponding 2D one. Could these just be a consequence of a "six degrees of separation", whereby both partners of a 3D-exclusive cooperation (A and B) are actually in a "true" cooperation with C and D respectively, and the fact that C and D themselves cooperate brings along A and B indirectly? Is there a way to decouple these effects in the analysis?

Response: We agree with the reviewer that the 3D-exclusive cooperation may be the result of indirect interactions. However, we cannot exclude the possibility that A and B may only interact directly in the 3D space but the interaction is not stable enough to be reliably detected by ChIP-seq experiments, i.e. pulling down B in A's ChIP-seq experiment. It is difficult to decouple these effects from solely computational analysis. What we have tried is to compare the percentage of 2D-exclusive, 3D-exclusive and 2D-3D cooperation edges supported by protein-protein interactions. We found that 11.5% and 11.5% of 3D-exclusive and 2D-3D cooperation edges, respectively, are coincident with protein-protein interactions, which is much higher than the 2D-exclusive edges (1.8%). Furthermore, 94.6%, 63.5% and 79.4% of 2D-exclusive, 3D-exclusive and 2D-3D cooperative DBP pairs are separated by one protein in the protein-protein interaction network, respectively. This observation suggests that our analysis did identify physical interactions in the 3D space and many 2D-exclusive ones may be formed through indirect interactions. We added this discussion to the text (Page 4).

	2D	3D	2D-3D
direct	1.8%	11.5%	11.5%
through one protein	94.6%	63.5%	79.4%

3. Similar to point 2, is there a way to test whether 2D AND 3D cooperation links to genotype variations are synergistic, compared to 2D only or 3D only?

Response: In the revision, we re-did this analysis and calculated the enrichments for each type of cooperation: 2D, 3D and 2D-3D. We did see a small increase of enrichment for the 2D-3D cooperation compared to 2D or 3D only, and the p-value seems more significant for 2D-3D cooperation (page 5, copied below).

"In Fig. 2g we showed that the vast majority of DBP cooperation are more enriched with disease associated GVs. We performed the Mann–Whitney U test to compare the significance level of enrichments of 1D-dominant, 3D-dominant and 1D-3D cooperative DBPs with that of uncooperative DBP pairs, the p-values are 1.6e-3, 5.2e-28 and 2.0e-45 respectively."

4. Is there a technical or biological explanation for the separation of the green and cyan communities? Do factors in one or other community "actively segregate"?

Response: These two communities share 3248 out of 5000 loci used in the GREAT analysis despite they are segregated in the network and the gene composition is largely different. We have revised the manuscript to clarify this point, as copied below.

" We then ranked genomic loci by the number of proteins bind to them and selected the top 5000 loci as input to GREAT analysis to search for enriched GO terms. We found that the green community is linked to mRNA metabolic process and translation related functions (Supplementary Fig. 2a). The same analysis showed that the cyan community is also enriched with similar GO terms. Interestingly, these two communities share 3248 out of the 5000 loci used in GREAT analysis despite they are segregated in the network (Supplementary Fig. 2b). Closer examination of these two communities revealed distinct protein composition. For instance, all the six histone modifications as well as RNA polymerase II belong to the green community, suggesting its pivotal roles in gene transcription. In contrast, the cyan community contains numerous proteins, including ESRRA, BRCA1, NRF1, ETS1 and STAT3, that are involved in the estrogen signaling pathway."

5. The supposed "rewiring" of DBP cliques across cell types is very interesting. Although exhaustive ChIP-seq datasets are not available to fully address this, is there a way to estimate how much of this rewiring can be accounted for by cell-specific factors participating in the complexes? If it's not 100%, what other factors could be responsible - a differential accessibility to cognate binding sites?

Response: We agree with the reviewer that there could be other factors contributing to the rewiring of DBP cliques, such as differential accessibility of chromatin or differential sequence preference studied in Arvey's paper[1]. We have discussed these possibilities in the revised manuscript. To assess the contribution of cofactors to such cell-type-specific binding, for each potential cofactor we used its ChIP-seq peaks to discriminate binding regions of STAT5A-CEBPB-PML in GM12878 and K562. The top 12 TFs that have best discriminative accuracy are RUNX3, TEAD4, TAL1, BCL11A, BATF, IRF4, PAX5, POU2F2, BCL3, GATA2, MYC and EBF1 (Supplemental Table 7). When the binding sites of these 12 TFs were used together to train a logistic regression model, we achieved a 100% accuracy rate for discriminating STAT5-CEBPB-

PML binding regions in two cell lines. Therefore, the cell-type-specific binding of this DBP clique can be explained by its partnership with different cofactors. Furthermore, we observed that all 12 TFs except MYC are differentially expressed in the two cell types (Supplemental Table 7), suggesting that the cell-type specific binding of this DBP clique is largely due to cell-type-specific expression of cofactors.

Other small modifications to consider:

1. On the first page, "mediator" is a poor choice of word, as it could be confused with components of the Mediator complex, which also participate in looping interactions.

Response: We have replaced "mediator" with "architectural protein", and rephrased the statements accordingly.

2. Although I sort of worked it out, a better and more formal explanation of the difference between a "community" and a "clique" would make the manuscript much easier to follow.

Response: We agree with the reviewer that the definition and the explanation of the community is not very clear in the original manuscript, which causes confusion. We have added detailed explanations for both communities and cliques in the revised manuscript, as copied below:

"Communities are groups of nodes in a network that are more densely connected internally than with the rest of the network. In other words, community structure is a partition or clustering of the nodes in a network."

"A clique is a complete sub-graph in which every pair of nodes is connected."

3. The manuscript is easy enough to follow, but should be proof-read by a native English speaker, as there are many small grammatical mistakes throughout.

Response: We have corrected many grammatical mistakes and made significant improvements to the overall writing.

4. It should be pointed out that a major limitation of this work is that it relies on very high-resolution Hi-C data, which is able to identify peaks over a sufficiently small window size. So far, this has only been achieved by the VERY EXPENSIVE work of Rao et al. (cited in the manuscript). Presumably, more datasets will follow, but it is a current bottleneck for the wider application of this work.

Response: We have discussed this limitation in Discussion, as copied below:

"One major limitation of this work is that it relies on high resolution Hi-C data which is very expensive. Furthermore, the chromatin loops used in this project are from the Rao et al. study[42]. Some weak loops may be missed as they used very stringent criteria to identify loops. However, as the sequencing technology rapidly evolves, these limitations will be overcome by the availability of more and more ChIP-seq and Hi-C data at even higher resolution."

Reviewer #3 (Remarks to the Author):

The authors perform a meta-analysis of 2D and 3D colocalization data to identify cooperating complexes that mediate 3D loop assembly. The idea is creative and the bioinformatics thought-provoking. Identification of combinatorial complexes that specify 3D loop formation would advance the field. Unfortunately, the authors claim correlative associations but fail either to demonstrate sufficient statistical rigor, or to test their claim that cooperation between the identified proteins indeed regulates 3D architecture. The authors should either more clearly and convincingly underwrite their bioinformatic meta-analysis, or carry out functional tests, or both.

Major concerns

Points 1-3 below concern the rigor of the bioinformatics, particularly the determinations of false discovery rates and the strength of the correlations between protein occupancy and loop assembly genome-wide:

1. The ROC curves presented in Fig. 1 only address simulations. The authors do not present an estimate of FDR using real data, so the reader cannot assess the validity of their analysis method.

Response: Because the ground truth of DBP cooperation is unknown, FDR cannot be calculated using the real data. Therefore, we used simulated data to assess the performance of our method. Note that the simulated networks came from random sampling of a real biological network and the data were generated using a biophysical model. Furthermore, we also generated a null model by random shuffling of DBP binding sites to represent uncooperative DBPs. When we applied our algorithm to this data set, we found zero cooperation using the default correlation cutoff (0.3), which shows our method has a low false discovery rate. We included this discussion in the revised text (see Methods).

2. In Fig. 3B, the "enrichment" value is not translated into a metric that allows evaluation of the statistical significance of a given clique. The authors should add to the table some measure of significance, the operational equivalent of a p-value.

Response: We have included p-values in Fig. 3B and the following was added to the text.

"We checked the percentage of shared peaks in the union of all DBP peaks and identified the loops that overlap with these shared peaks. As a comparison, for each k-component DBP clique we randomly selected k DBPs and did the same analysis. We took 50000 samples and used them as a null model for enrichment and p-value calculation."

3. The strength of the correlation between 2D colocalization and 3D association for the bulk population of identified sites is not clear. Although the chosen examples in Figures 2E, 2F, and 3D are convincing, a global visualization that displays all identified sites would help convince the reader that the thousands or so sites are indeed enriched.

Response: We have updated Figures 2E, 2F and 3D so that much larger genomic regions are displayed. Note that these figures are only visualization of cooperative DBPs in the genome browser and quantitative measurements of the co-occurrence between DBPs are shown in Fig. 3B.

Points 4-7 below concern the functional significance of the correlations observed here:

4. The authors define protein "communities" based on network analysis and assign functions based on GO ontology terms. It seems likely that these assemblies include many enhancer-promoter interactions that may regulate the genes in these processes. Is it possible to build on a functional correlation by identifying contacts with promoters identifying enriched GO ontology?

Response: Following the reviewer's suggestion, we performed the GO term analysis focusing on promoters (-5 kb to +1 kb from transcription start sites) contacting with binding sites of DBP complexes. The results are summarized in the table below. The GO terms found in the new analysis are relatively more general compared to the old analysis, such as "metabolic process" in the new analysis compared to "mRNA metabolic process" in the old analysis for the green community. The GREAT analysis considers "neighbor" genes in the linear genome to the DBP binding sites but did not consider the distal promoters that are in spatial contact with these binding sites. It would be a great idea for the GREAT developers to incorporate these 3D information into GREAT analysis.

	Community	Enriched GO terms
new analysis	Green	metabolic process cellular metabolic process organic substance metabolic process primary metabolic process cellular macromolecule metabolic process macromolecule metabolic process nitrogen compound metabolic process heterocycle metabolic process cellular aromatic compound metabolic process cellular nitrogen compound metabolic process
Original analysis	Green	mRNA metabolic process translation nuclear-transcribed mRNA catabolic process mRNA catabolic process nuclear-transcribed mRNA catabolic process, nonsense-mediated decay viral gene expression macromolecular complex disassembly protein complex disassembly translational termination RNA catabolic process
new analysis	Red	cellular macromolecule metabolic process macromolecule metabolic process nucleic acid metabolic process cellular response to organic substance regulation of immune system process positive regulation of protein metabolic process organelle organization DNA metabolic process regulation of apoptotic process regulation of programmed cell death

original analysis	Red	immune response positive regulation of immune system process leukocyte activation regulation of immune response cell activation lymphocyte activation positive regulation of immune response regulation of leukocyte activation regulation of cell activation regulation of lymphocyte activation
new analysis	Cyan	cellular process metabolic process cellular metabolic process organic substance metabolic process primary metabolic process cellular macromolecule metabolic process macromolecule metabolic process nitrogen compound metabolic process cellular nitrogen compound metabolic process cellular aromatic compound metabolic process
original analysis	Cyan	mRNA metabolic process translation nuclear-transcribed mRNA catabolic process mRNA catabolic process translational initiation RNA catabolic process nuclear-transcribed mRNA catabolic process, nonsense-mediated decay translational termination translational elongation viral gene expression

5. Two new potential structural complexes are inferred, PML-FOXM1-etc. and MAX-MAZ-etc. Functional tests of these presumptive complexes are needed. Does knockdown of one component affect 2D binding of another? How many components are necessary to maintain the 3D loop?

6. The inferred cell type-specific complexes, such as those containing STAT5, require functional tests. Such experiments could present a wonderful opportunity to demonstrate that combinatorial control imparts context-specific 3D function.

Response: We totally agree with the reviewer that these experiments proposed by the reviewer in both point 5 and 6 are very interesting, and would significantly boost the importance of this work. However, the main goal of this study is to develop a novel computational method that can systematically identify cooperative proteins that possibly mediate chromatin looping by integrating TF binding and Hi-C data. The newly identified protein complexes can guide biologists for the follow up mechanistic studies.

The mechanistic studies will involve extensive effort and time, which we strongly feel is out of the scope of this study. For example, as suggested by the reviewer, knockdown of a single component may not be sufficient to disrupt the loops. A rigorous study should knock down every

individual component, every pair of components, and so on, and examine the binding of the other components in the complex. Furthermore, one needs to carry out Hi-C experiments to obtain a global view of the loop disruption after each knockdown experiment. These experiments are way beyond validation of our computational predictions and thus out of the scope of this computational paper.

We also would like to emphasize that the goal of our method is to extract the existing cooperation between DBPs detected by the ChIP-seq and Hi-C measurements. We did not predict them. Uncovering the mechanisms underlying these cooperation through functional tests would be more appropriate for a separate paper and indeed we have started working with our collaborators to set up live imaging experiments to monitor the loop formation process mediated by these complexes in vivo.

7. The assertion that the GGM network discovers direct interactions overreaches, because protein-protein interaction databases include findings from procedures that detect indirect associations, e.g., immunoprecipitation followed by mass spec. Follow up studies, such as in vitro binary interaction data, and mutations that abrogate the interaction in vitro and in vivo, would be needed to conclude direct protein-protein interaction. The authors should to soften their claims about the directness of the inferred interactions.

Response: We agree with the reviewer and have revised the manuscript accordingly.

References:

1. Arvey, A., et al., *Sequence and chromatin determinants of cell-type-specific transcription factor binding*. *Genome Res*, 2012. **22**(9): p. 1723-34.

Reviewer #1 (Remarks to the Author):

The authors have addressed all my concerns. I recommend the publication of this work.

Reviewer #2 (Remarks to the Author):

The authors have satisfactorily replied to all comments, and I support the manuscript for publication.

Reviewer #3 (Remarks to the Author):

The authors have successfully extracted new information from the large data sets they analyzed, and addressed thoughtfully most of my concerns. I am comfortable with acceptance of this revised version.

We thank the referees for their useful comments and recommendations of publication of this manuscript!